# Separating random and deterministic sources of computational noise in explore-exploit decisions

Siyu Wang[1]*, Robert C. Wilson[1,2,3,4,5]*

**1** Department of Psychology, University of Arizona, Tucson, Arizona, United States of America,
**2** Neuroscience and Physiological Sciences Graduate Interdisciplinary Program, University of Arizona, Tucson, Arizona, United States of America, **3** Cognitive Science Program, University of Arizona, Tucson, Arizona, United States of America, **4** School of Psychology, Georgia Institute of Technology, Atlanta, Georgia, United States of America, **5** Center of Excellence for Computational Cognition, Georgia Institute of Technology, Atlanta, Georgia, United States of America

\* wangxsiyu@gmail.com (SW); bob.wilson@gatech.edu (RCW)

## Abstract

Human decision making is inherently variable. While this variability is often seen as a sign of suboptimal behavior, both theoretical work in machine learning and empirical human studies suggest that variability can actually be adaptive. An example arises when we must choose between exploring unknown options or exploiting options we know well. A little randomness in these 'explore-exploit' decisions is remarkably effective as it can encourage us to explore options we might otherwise ignore. In line with this idea, several studies have found evidence that people increase their behavioral variability when it is valuable to explore. A key question, however, is whether this variability in so-called 'random exploration' is actually random. That is, is random exploration driven by stochastic processes in the brain or by some unobserved deterministic process that we have failed to account for when measuring behavioral variability? By designing an explore-exploit task in which, unbeknownst to them, participants are presented with the exact same choice twice, we provide a partial answer to this question. By modeling behavior in this task, we were able to estimate a lower bound on the amount of variability that is deterministically driven by the stimulus and an upper bound on the amount of variability that is random. Using this approach, we find evidence that at least 14% of the variability in random exploration in our studied task can be accounted for by deterministic processing of the stimulus. Conversely, this suggests that up to 86% of the variability is truly 'random', although it is still possible that this variability is driven by deterministic factors not related to the stimulus. Finally, our results suggest that both deterministic and random sources of variability change proportionally to each other as the value of exploration increases, suggesting that a common noise gating mechanism may be at play in random exploration.

**Data availability statement:** Behavioral data is available at https://doi.org/10.6084/m9.figshare.31396443. MATLAB codes used to create all analyses and figures are available at https://github.com/wangxsiyu/Paper_PlosCB_2026_RandomDeterministicNoise.git.

**Funding:** This work is supported by National Institute on Aging grants awarded to RC (R01 AG061888, R56 AG061888). The funders had no role in study design, data collection and analysis, decision to publish, or preparation of the manuscript.

## Author summary

Human decisions often seem random. Even simple decisions like what food to order at a restaurant can be difficult to predict ahead of time. This randomness in our decisions can be beneficial, effectively allowing us to explore new options. One outstanding question is where the randomness in our decisions comes from. Sometimes, our seemingly random decisions are driven by predictable external factors, like what the guests at the next table order could influence what we order. Other times, our decisions are not driven by external factors but are instead made by random thoughts within our brain. In this work, we developed a computational method that quantifies the extent to which the apparent randomness in our decisions can be explained by deterministic sources of variability in the external stimuli, or random variability unexplained by the stimuli. We found evidence that randomness in exploratory decisions can be explained by both random (up to 86%) and deterministic (more than 14%) sources of variability. Moreover, our results suggest that both sources of variability are adaptive, which enables humans to explore more when it is more beneficial to explore. The joint adaptation of random and deterministic noises also suggests a common noise-gating mechanism for exploration.

## Introduction

Imagine trying to decide where to go to dinner on a date. You can go to your favorite restaurant, the one you both really enjoy and always go to, or you can try a new restaurant that you know nothing about. Such decisions, in which we must choose between a well-known 'exploit' option and a lesser known 'explore' option, are known as explore-exploit decisions. From a theoretical perspective, making optimal explore-exploit choices, i.e., choices that maximize long-term reward, is computationally intractable in most cases [1,2]. In part because of this computational complexity, there is considerable interest in how humans and animals solve the explore-exploit dilemma in practice [3–5].

One particularly effective strategy for solving the explore-exploit dilemma is choice randomization [6–8], also known as random exploration. In this strategy, high value 'exploit' options are not always chosen and exploratory choices are sometimes made by chance. From a modeling perspective, random exploration works by adding 'decision noise' to the value of the options such that sub-optimal exploratory options can sometimes have a higher total score (i.e., value + noise) than the exploit option and get chosen. Such random exploration, is surprisingly effective and, if implemented correctly, can come close to optimal performance [6,8–10].

It has recently been shown that humans appear to use random exploration and can increase decision noise when it is more beneficial to explore [11,12], as has also been suggested in computational models of animal behavior [13,14]. In one of these tasks, known as the Horizon Task [12], the key manipulation is the horizon condition,

i.e., the number of decisions remaining for the participant to make. Increasing the horizon makes exploration more valuable as there is more time to use the information gained by exploration to maximize future rewards. For example, if you are leaving town tomorrow (short horizon), you will probably exploit the restaurant you know and love, but if you are in town for a while (long horizon), you will be more likely to explore the new restaurant. Using such a horizon manipulation it has been shown that people's behavior is more variable in long horizons than short horizons, suggesting that they use adaptive decision noise to solve the explore-exploit dilemma [12].

One limitation of this previous research, however, is that it is difficult to tell whether what we have called 'decision noise' actually reflect a noise process. From a modeling perspective, decision noise as defined in previous research essentially quantifies the extent to which behavior cannot be explained by a computational model. A missing deterministic component from the model could give rise to variability in behavior that might appear to be random noise. For example, in the restaurant example, my usual preference for one restaurant or another may be overruled if I see an ex romantic partner going into one of them. Avoiding an ex is a deterministic process, but if we fail to take the ex's presence into account as scientists modeling the decision, then over a series of such decisions where the ex is present or not, we would mistakenly attribute the ensuing 'variability' in choice to randomness. To dissociate a missing deterministic component from a true random process, choice consistency between repeated decisions can be utilized to decompose behavioral variability into predictable deterministic components and unpredictable random components [15–18].

In this paper, we investigate the extent to which the apparent randomness in random exploration can be explained by deterministic processing of the stimulus (which we refer to as 'deterministic noise') vs other processes, including deterministic processing that is unrelated to the stimuli as well as truly stochastic processes (which we refer to as 'random noise'). To distinguish between these two types of noise, we modify the Horizon Task [12] to have people face the exact same explore-exploit choice twice. If the decision is a purely deterministic function of the stimulus (i.e., decision noise is purely deterministic noise), then people's choices should be identical for both decisions, since the stimulus is the same both times. Conversely, if the decision is a purely random function of the stimulus (i.e., decision noise is purely random noise), then people's choices will be different 50% of the time, since the random noise is different each time. In between these two extremes of purely deterministic and purely random drivers of behavioral variability, the extent to which people's decisions are consistent between the two decisions can be used to estimate the amount of deterministic and random noise.

In the following, we analyze behavior on the repeated decisions version of the Horizon Task in both a model-free and model-based manner. Our model-free analysis estimates the extent to which people's behavior is consistent across repeated versions of the same decision. By measuring how this choice consistency changes as a function of horizon, this model-free analysis offers qualitative insight into the extent to which behavioral variability is driven by deterministic vs random noise. Our model-based analysis uses a computational model of the explore-exploit decision in the Horizon Task that incorporates both noise processes. By fitting this model to the behavioral data, this model-based analysis allows us to quantify the relative size of the two sources of noise and how they change in the service of exploration.

## Results

### The repeated-games horizon task

We used a modified version of the 'Horizon Task' [12] to show the influence of stimulus-driven 'deterministic noise' vs non-stimulus-driven 'random noise' in explore-exploit decisions (Fig 1). In this task, participants make a series of choices between two slot machines, or 'one-armed bandits', that pay out probabilistic rewards. They are asked to choose between the two bandits to maximize the total rewards. One bandit always has a higher mean payout than the other. Participants need to try each bandit a few times to learn about the distribution of payout from that bandit. Because they are initially unsure as to the mean payoff of each bandit, this task requires that participants carefully balance exploration of the lesser known bandit with exploitation of the better known bandit to maximize their overall rewards.

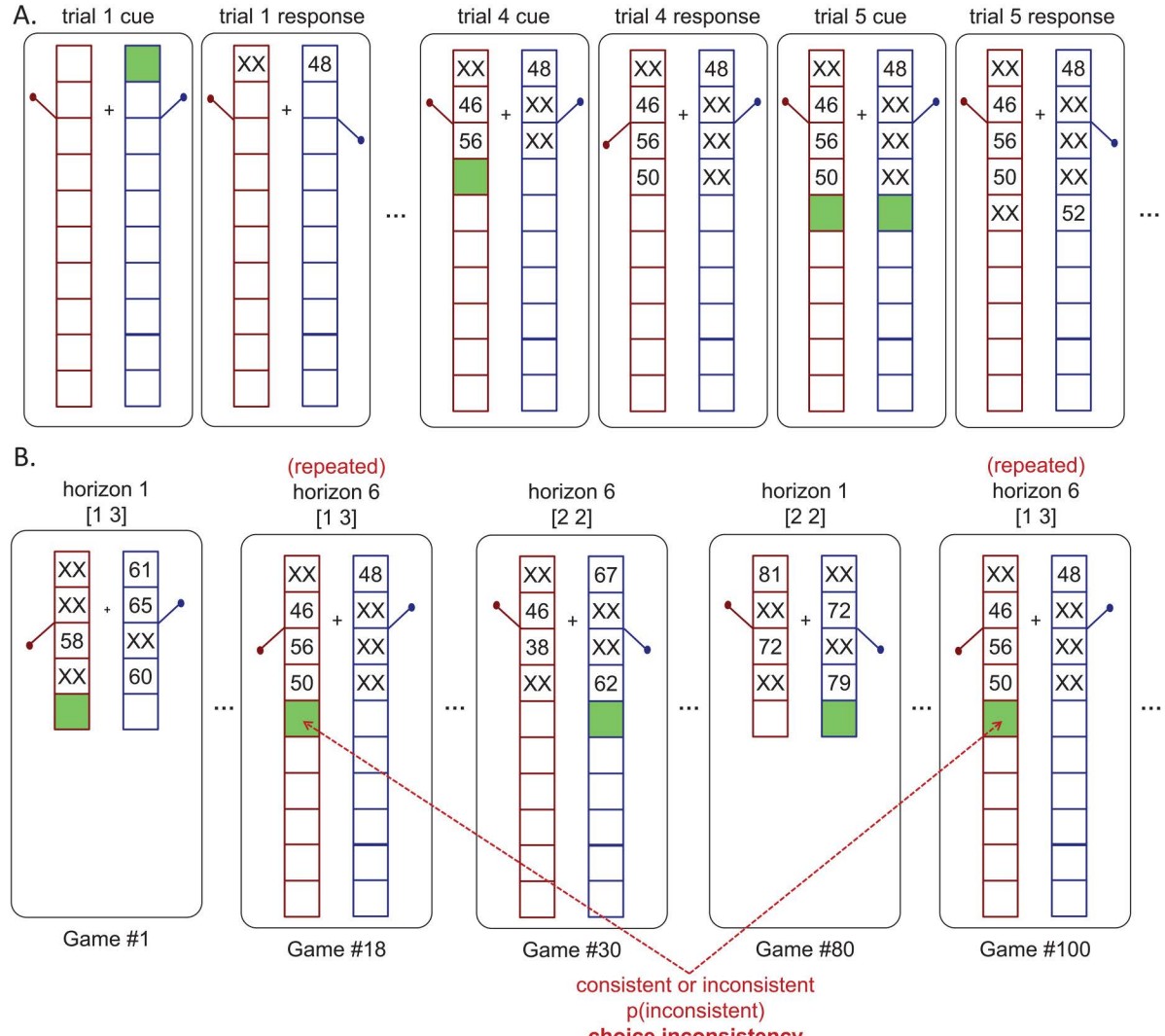

**Fig 1. Schematic of the experiment. (A)** Dynamics of an example horizon 6 game. Here the first four trials are forced trials in which participants are instructed which option to play. After the forced trials, participants are free to choose between the two options for the remainder of the game. **(B)** Example repeated games over the course of the experiment. On average, participants play more than 150 such games, with varying horizon (1 vs 6), uncertainty condition ([1 3] vs [2 2]) and observed rewards. In addition, all games are repeated (as Game 18 and 100 are here) such that participants will be faced with the exact same pattern of forced trials and exact same outcomes from those forced trials twice within each experiment. These repeated games allow us to compute the relative contribution of deterministic and random noise by analyzing the extent to which choices are *consistent* across the repeated games.

The task is organized in games (Fig 1A). The mean payout of the two bandits are held fixed within a game and reset between games. Each game consists of either 5 or 10 trials. The first four trials of each game are 'forced-choice' trials. In the first four trials, participants are instructed about which bandit to choose, this allows us to manipulate what information from both bandits participants receive before they make their first free choice between the two bandits. From the 5th trial, participants make free choices between the two bandits. Participants have either 1 or 6 free choices to make.

The Horizon Task has two key features that together allow it to quantify explore-exploit behavior. The first of these features is the time horizon — the number of decisions participants will make in the future. By changing this horizon from

short (1 free-choice trial) to long (6 free-choice trials), the Horizon Task allows us to control the relative value of exploration and exploitation. Just like the restaurant example in the introduction, when the horizon is short, participants should be more likely to exploit the option they believe to be best, because this leads to the highest payoff in the short term. Conversely, when the horizon is long, participants should be more likely to explore at first, because this allows them to gather information to make better choices later on. By contrasting behavior between short and long horizon conditions *on the very first free-choice trial*, when all else is equal, the Horizon Task allows us to quantify how behavior changes, when it is more valuable to explore.

The second key feature of the Horizon Task are the 4 forced-choice trials at the start of each game that allow us to control exactly what participants know about the two bandits before they make their choice. In these forced-choice trials, participants are instructed which of the bandits to play allowing us to control how much information they have about each of the options. The forced-choice trials are used to set up one of two information conditions: an 'unequal information' or [1 3] condition, in which participants play one bandit once and the other three times, and an 'equal information' or [2 2] condition, in which participants play both bandits twice.

Relative to the original Horizon Task, the key modification in this paper is to give people 'repeated games' (Fig 1B), in which they see the exact same set of forced-choice plays twice in two separate games separated by several minutes in time so as to avoid detection. By repeating the forced-choice plays for each game twice, we can set up a situation where (unbeknownst to the participants) they are faced with the exact same explore-exploit choice, with the exact same stimuli twice. Thus, if their behavior is a deterministic function of the stimuli, then they will make the same decision in both games and their choices will be consistent. Conversely, if their behavior is not driven by a deterministic function of the stimulus, then their choices on the repeated games will be inconsistent some fraction of the time. The extent to which participants' choices are consistent on the repeated versions of the games allow us to quantify the extent to which the variability in their behavior was driven by a deterministic process vs a random noise process.

## Both behavioral variability and information seeking increase with horizon

Before discussing the results for repeated games, we first confirm that the basic behavior in this task is consistent with our previously reported results using both a model-free and model-based approach [12]. In both analyses, we focus on just the first free-choice trial in each game, where the only thing that differs between the horizon conditions is the number of choices that participants will make in the future. Subsequent choices in Horizon 6 games were not analyzed.

## Model-free analysis

In the model-free analysis, we quantify random and directed exploration using simple choice probabilities. Random exploration is quantified as the probability of choosing the option that has the lower average payout in the forced-choice plays in the equal, or [2 2], condition, $p(\text{low mean})$. The idea here is that, in the equal condition, the optimal strategy is to compute the mean payout for each bandit from the forced-choice plays and then always choose the option with the highest mean. When participants do not choose the option with the higher mean, the assumption is that this is due to some kind of 'decision noise', making the probability of choosing the low mean option a measure of behavioral variability. In this view, random exploration corresponds to an increase in $p(\text{low mean})$ with horizon, which is exactly what we see in the data (Fig 2A; $t(64) = 7.99$, $p < 0.001$).

Directed exploration is quantified as the probability of choosing the more informative option $p(\text{high info})$ in the unequal, or [1 3], condition. The more informative option is the option played once during the forced-choice plays as choosing this option gives relatively more information (doubling the number of samples from 1 to 2) than choosing the option played three times (only increasing the number of sample by a third, from 3 to 4). In this view, directed exploration corresponds to an increase in $p(\text{high info})$ with horizon, which is exactly what we see in the data (Fig 2B; $t(64) = 6.92$, $p < 0.001$).

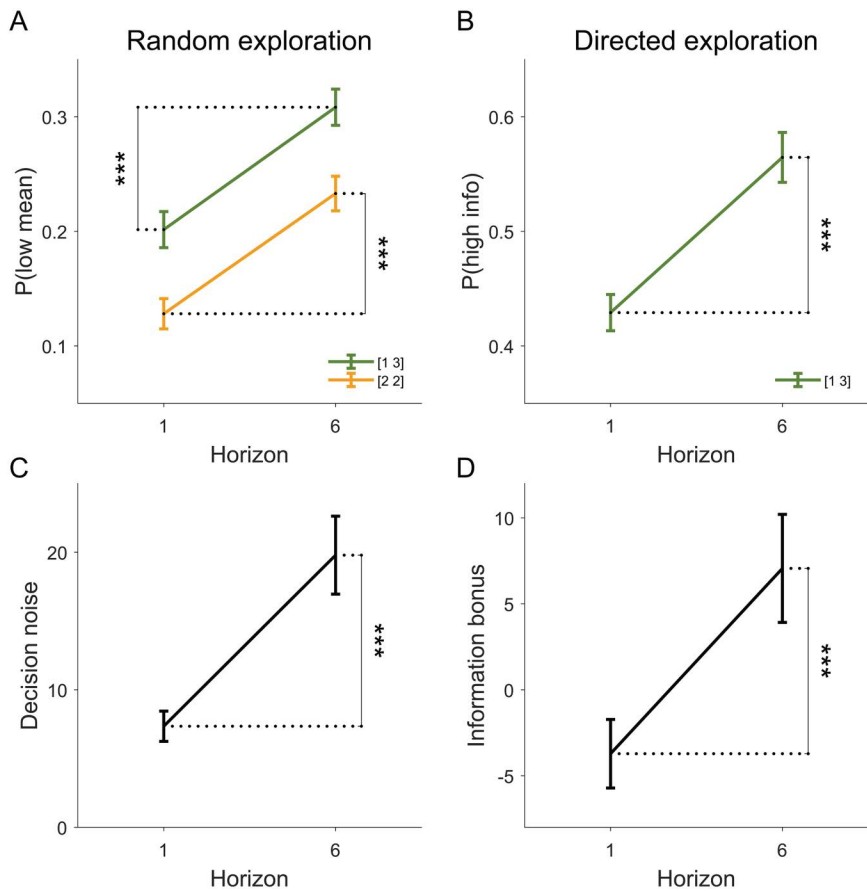

**Fig 2. Replication of previous findings that people use both random and directed exploration in this task.** (A) model-free measure of behavioral variability, $p$(low mean), increases with horizon. (B) model-free measure of information seeking, $p$(high info), increases with horizon. (C) model-based measure of behavioral variability, decision noise $\sigma$, increases with horizon. (D) model-based measure of information seeking, information bonus $A$, increases with horizon.

**Model-based analysis.** Another approach to understanding behavior in the Horizon Task is to use a computational model [12]. In this case, we model participants' choices on the first free-choice trial by assuming they make decisions by computing the difference in value (or utility) $\Delta Q$ between the right and left options, choosing right when $\Delta Q > 0$ and left otherwise. Specifically, we write

$$\Delta Q = \Delta R + A \Delta I + b + n \tag{1}$$

where, the experimentally controlled variables are $\Delta R = R_{right} - R_{left}$, the difference between the mean of rewards shown on the forced-choice trials, and $\Delta I$, the difference in information available for playing the two options on the first free-choice trial. For simplicity, and because information is manipulated categorically in the Horizon Task, we define $\Delta I$ to be + 1 if one reward is drawn from the right option and three are drawn from the left in the [1 3] condition, -1 if one from the left and three from the right in the [1 3] condition, and in [2 2] condition, $\Delta I$ is 0.

Here, $n$ denotes decision noise, which, in this version of the model is a combination of deterministic and random noise. $n$ is assumed to come from a logistic distribution with mean 0 and standard deviations $\sigma$.

The free parameters of this model are: the information bonus $A$, which controls the level of directed exploration; the noise standard deviation, $\sigma$, which controls the level of random exploration, and the spatial bias, $b$, which determines the extent to which participants prefer the option on the right. These free parameters are fit separately for each participant in each horizon condition, allowing us to test whether directed and random exploration increase with horizon. Consistent with previous research, we find that this is indeed the case (Fig 2C; t(64) = 5.35, p<0.001. Fig 2D; t(64) = 3.54, p<0.001).

Taken together, our model-free and model-based analyses agree with previous findings showing increased behavioral variability and increased information seeking in the long horizon condition, consistent with humans using random and directed exploration (Figs 2 and S1). However, for random exploration, this previous analysis cannot distinguish between deterministic and random sources of noise. For this we analyze the extent to which people's choices are consistent on the repeated games.

### Model-free analysis of repeated games suggests that random exploration involves both random and deterministic noise

Next we asked whether participants' choices were consistent or inconsistent in the two repetitions of each game. The idea behind this measure is that purely deterministic noise should lead to consistent choices as the deterministic stimulus is identical both times. Conversely, if choice is not entirely driven by a deterministic process and is also driven by random noise, participants' choices should be more inconsistent across the repetitions of the game. Moreover, if decision noise is purely random noise, meaning there is no unobserved deterministic process, we will show that we can actually predict the expected level of choice inconsistencies across repetitions of games by accounting for the known deterministic processes and assuming that the random noise process is independent in repetitions of the game.

To quantify choice inconsistency we computed the frequency with which participants made different responses for pairs of repeated games (Figs 3 and S2). Using this measure we found that participants made inconsistent choices in both the unequal ([1 3]) and equal ([2 2]) information conditions (p(inconsistent) > 0), suggesting that not all of the noise was stimulus driven. In addition, we found that choice inconsistency was higher in horizon 6 than in horizon 1 for both [1 3] and [2 2] condition (For [1 3] condition, t(64) = 5.41, p<0.001; for [2 2] condition, t(64) = 6.26, p<0.001), suggesting that at least some of the horizon dependent noise is not a deterministic function of the stimulus, but rather random noise.

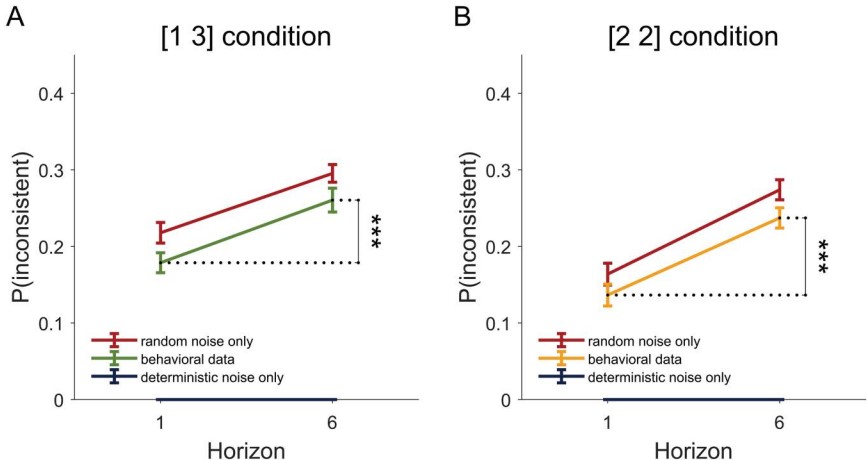

**Fig 3. Model-free analysis suggests that both deterministic and random noise contribute to the choice variability in random exploration.** For both the [1 3] (A) and [2 2] (B) condition, people show greater choice inconsistency in horizon 6 than horizon 1. However, the extent to which their choices are inconsistent lies between what is predicted by purely deterministic and random noise, suggesting that both noise sources influence the decision.

To gain more quantitative insight into these results, we computed theoretical values for the choice inconsistency for the purely deterministic and purely random noise cases. For purely deterministic noise this computation is simple because people should make the exact same decisions each time in repeated games, meaning that $p$(inconsistent) = 0 in this case. For purely random noise, the two games should be treated independently. Since repeated decisions mean that participants either choose the low-mean option twice, or choose the high-mean option twice, we could predict the choice inconsistency, $p$(inconsistent), based on the probability of choosing the low mean option, $p$(low mean), as

$$p(\text{consistent}) = p(\text{low mean})^2 + p(\text{high mean})^2$$
$$= p(\text{low mean})^2 + (1 - p(\text{low mean}))^2$$
$$\text{hence,} \quad p(\text{inconsistent}) = 1 - p(\text{consistent}) = 2p(\text{low mean})(1 - p(\text{low mean}))$$

Furthermore, to account for the fact that $p$(low mean) is a function of reward difference $\Delta R$ between the two bandits and the information condition $I$, we estimated the conditional probability:

$$p(\text{inconsistent}|\Delta R, I) = 2p(\text{low mean}|\Delta R, I)(1 - p(\text{low mean}|\Delta R, I))$$

Then based on the likelihood that each condition ($\Delta R$ and $I$) occurs in the task $\rho(\Delta R, I)$, we have

$$p(\text{inconsistent}) = \sum_{\Delta R, I} \rho(\Delta R, I)p(\text{inconsistent}|\Delta R, I)$$

As shown in Fig 3, people's behavior falls in between the pure deterministic noise prediction and the pure random noise prediction. Specifically, behavior is different from the pure random noise prediction in both the [1 3] condition ($t(64) = 4.83$, $p < 0.001$ for horizon 1, $t(64) = 3.12$ $p = 0.003$ for horizon 6) and the [2 2] condition ($t(64) = 3.92$, $p < 0.001$ for horizon 1, $t(64) = 3.71$, $p < 0.001$ for horizon 6). Likewise, behavior is different from pure deterministic noise prediction in both the [1 3] condition ($t(64) = 13.72$, $p < 0.001$ for horizon 1, $t(64) = 16.71$, $p < 0.001$ for horizon 6) and the [2 2] condition ($t(64) = 9.55$, $p < 0.001$ for horizon 1, $t(64) = 17.93$, $p < 0.001$ for horizon 6). As a negative control of our method for estimating $p$(inconsistent) for purely random noise, we simulated choices using a decision model that only includes random noise (Equation 2), and found that $p$(inconsistent) in this simulated data is not different from our pure random noise prediction in all horizon and uncertainty conditions ($p > 0.05$, S3 Fig). Together, our results suggest that both random noise and deterministic noise contribute to the choice variability in random exploration. However, the relative contribution from each of these types of noise, as well as how each type of noise changes with horizon, are difficult to discern.

## Model-based analysis provides a lower-bound estimate of deterministic noise and an upper-bound estimate of random noise

To more precisely quantify the contribution of deterministic noise and random noise, we turned to model fitting. We modeled behavior on the first free choice of the Horizon Task using a version of the logistic choice model (Equation 1) that was modified to differentiate between components of the noise that are deterministically driven by the stimulus ('deterministic noise') and components of the noise that are not deterministically driven by the stimulus ('random noise'). In particular, we assume that in repeated games, the value of stimulus-driven deterministic noise is frozen whereas random noise is drawn independently both times.

**Overview of model.** To model participants' choices on the first free-choice trial, we use a modified version of Equation 1.

$$\Delta Q = \Delta R + A\Delta I + b + n_{det} + n_{ran} \tag{2}$$

where, as before $\Delta R$, is the the difference in mean rewards shown on the forced-choice trials, $\Delta I$, is the difference in information, $A$ is the information bonus, and $b$ is the spatial bias. New in Equation 2 are the terms $n_{det}$ and $n_{ran}$. $n_{det}$ denotes the deterministic noise, which is identical on the repeat versions of each game; and $n_{ran}$ denotes random noise, which is uncorrelated between repeated plays and changes every game. $n_{det}$ and $n_{ran}$ are assumed to come from logistic distributions with mean 0, and standard deviations $\sigma_{det}$ and $\sigma_{ran}$.

For each pair of repeated games, the set of forced-choice trials are exactly the same, so the deterministic noise, $n_{det}$, should be the same while the random noise, $n_{ran}$ may be different. This is exactly how we distinguish deterministic noise from random noise. In symbolic terms, for repeated games $i$ and $j$, $n_{det}^i = n_{det}^j$ and $n_{ran}^i \neq n_{ran}^j$. While an increase in either random or deterministic noises could lead to higher $p$(low mean), an increase in random noise predicts higher $p$(inconsistent) while an increase in deterministic noise predicts lower $p$(inconsistent).

We used hierarchical Bayesian analysis to fit the parameters of the model (see Fig 8 for a graphical representation of the model in the style of [19]). In particular, we fit values of the information bonus $A$, spatial bias $b$, variance of random noise $\sigma_{ran}^2$, and variance of deterministic noise $\sigma_{det}^2$ for each participant in each horizon. Model fitting was performed using the MATJAGS and JAGS software [20,21] with full details given in the Methods.

In addition to the model presented here, we also considered and fit a series of reduced and alternative models to the data. This includes reduced models that assume only deterministic or random noises (Table A in S1 Text). We also considered a model with a non-categorical definition of $\Delta I$ that $\Delta I$ is defined to be the difference of the variances of rewards shown on the forced-choice trials between the two bandits. Lastly, we fit models in which the standard deviation of random and deterministic noises $\sigma_{det}$ and $\sigma_{ran}$ are estimated separately for the [1 3] and [2 2] information conditions. Additional details on these model variants are presented in section 2.1 in S1 Text.

**Model validation.** To be sure that our fit parameter values were meaningful and to understand the limits of our model, we evaluated our model extensively using simulated data. Qualitatively, we showed that our way of modeling deterministic noise is capable of capturing known deterministic processes intentionally omitted from the full model (S4 Fig). Quantitatively, we evaluated whether deterministic and random noise can be identified under ideal conditions where the behavior is generated by the model with known parameters (S5-S10 Figs). Full details are presented in sections 1.1 - 1.4 in S1 Text.

In this section we focus on our results for parameter recovery [22]. In a parameter recover analysis, behavioral data is simulated by the model with known parameters and then this simulated behavioral data is fit with the model to quantify the extent to which fit parameters match the input simulated parameters — that is, whether the simulated parameters can be recovered.

Parameter recover in this task was good for this model (Figs 4, S7 and S8), with fit values for $\sigma_{ran}$ and $\sigma_{det}$ showing strong correlations with their simulated values in both horizon (H) conditions (For $\sigma_{ran}$, $R = 0.91 (H = 1)$ and $0.84 (H = 6), p < 0.001$, For $\sigma_{det}$, $R = 0.60 (H = 1)$ and $0.59 (H = 6), p < 0.001$). However, while the relationship was near perfect for random noise ($\frac{\text{Recovered } \sigma_{ran}}{\text{Simulated } \sigma_{ran}} = 1.01$), there was a systematic bias to underestimate the level of deterministic noise by about 32% ($\frac{\text{Recovered } \sigma_{det}}{\text{Simulated } \sigma_{det}} = 0.68$). Despite this underestimation of deterministic noise in both horizon conditions, the difference in deterministic noise between horizons is much better captured (see section 1.2 in S1 Text). This is because the underestimation of deterministic noise is partially canceled out when the difference is taken between horizon conditions. In addition, we see better parameter recovery for random noise than deterministic noise. This is likely because we effectively have half as many trials for deterministic noise. In particular, while we generate two samples of random noise for each repeated game pair, we only generate one sample of deterministic noise, which by definition is the same in both of the repeated games.

In addition to the conventional subject-level parameter recovery analysis presented here, we also performed parameter recovery analysis that examined how faithful the full posterior distribution of group-level parameters can be recovered in

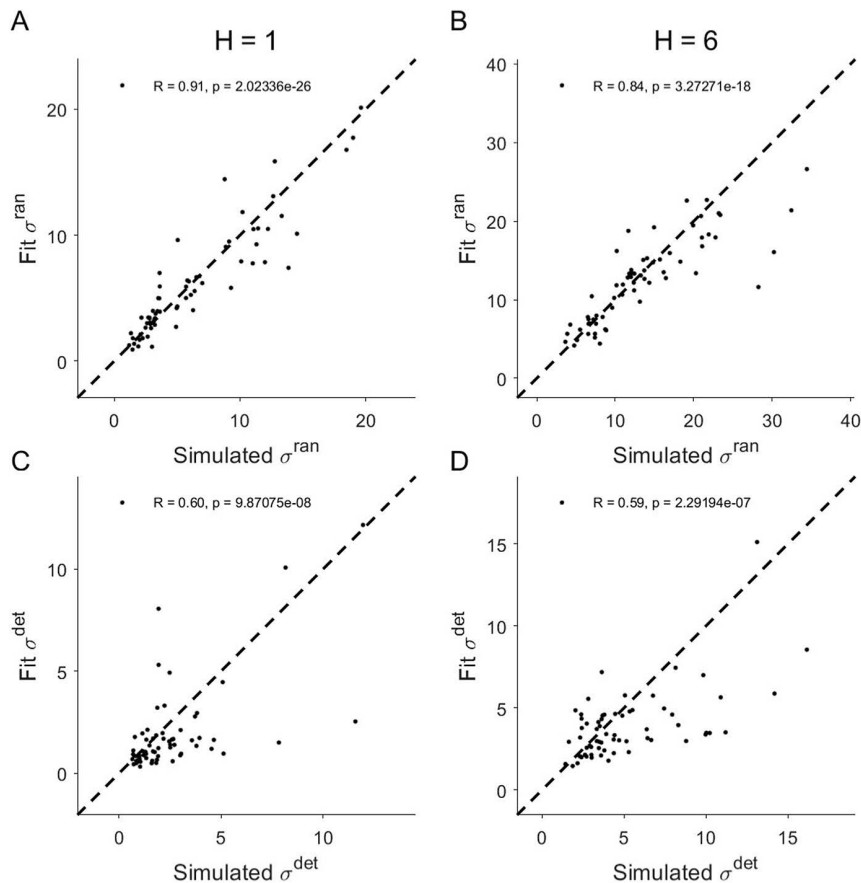

**Fig 4. Parameter recovery analysis for random (A,B) and deterministic (C,D) noises in the two horizons.**

simulated data (S5, S6, S9, and S10 Figs). Full details of these additional analyses are presented in sections 1.2 and 1.4 in S1 Text.

Overall, we were able to detect both deterministic and random noises using our model. Because random noise is modeled as non-stimulus-driven noise, it can reflect both true stochastic random noise and possible deterministic noises which do not depend on the stimuli. Thus conceptually our random noise estimate provides an upper bound for the true 'random noise' induced by intrinsic stochastic processes in the brain. Thus, our model provides a lower bound for deterministic noise and an upper bound for random noise.

**Model-based results.** Posterior distributions over the group-level means of the deterministic and random noise standard deviation $\sigma_{det}$ and $\sigma_{ran}$ are shown in Figs 5 and S11. Consistent with our model-free results, we see that both random and deterministic noise are non-zero. Numerically, random noise is about 2–3 times larger than the deterministic noise. By computing the posterior distribution of $\sigma_{det}^2/(\sigma_{det}^2 + \sigma_{ran}^2)$, our data suggests that 14.25% of the variability in random exploration is accounted for by deterministic noise ([4.90%, 28.81%], 95% CI). In addition, we find that both random and deterministic noise increase with horizon. This increase was larger for random noise (mean = 7.13, 100% of samples showed an increase in random noise with horizon) than deterministic noise (mean = 2.59, 98.64% of samples showed an increase in deterministic noise with horizon). But intriguingly, the relative increase in both types of noise was similar (Fig 6). That is, when we compute the relative increase in deterministic noise with horizon, $\sigma_{horizon6}^{det}/\sigma_{horizon1}^{det}$,

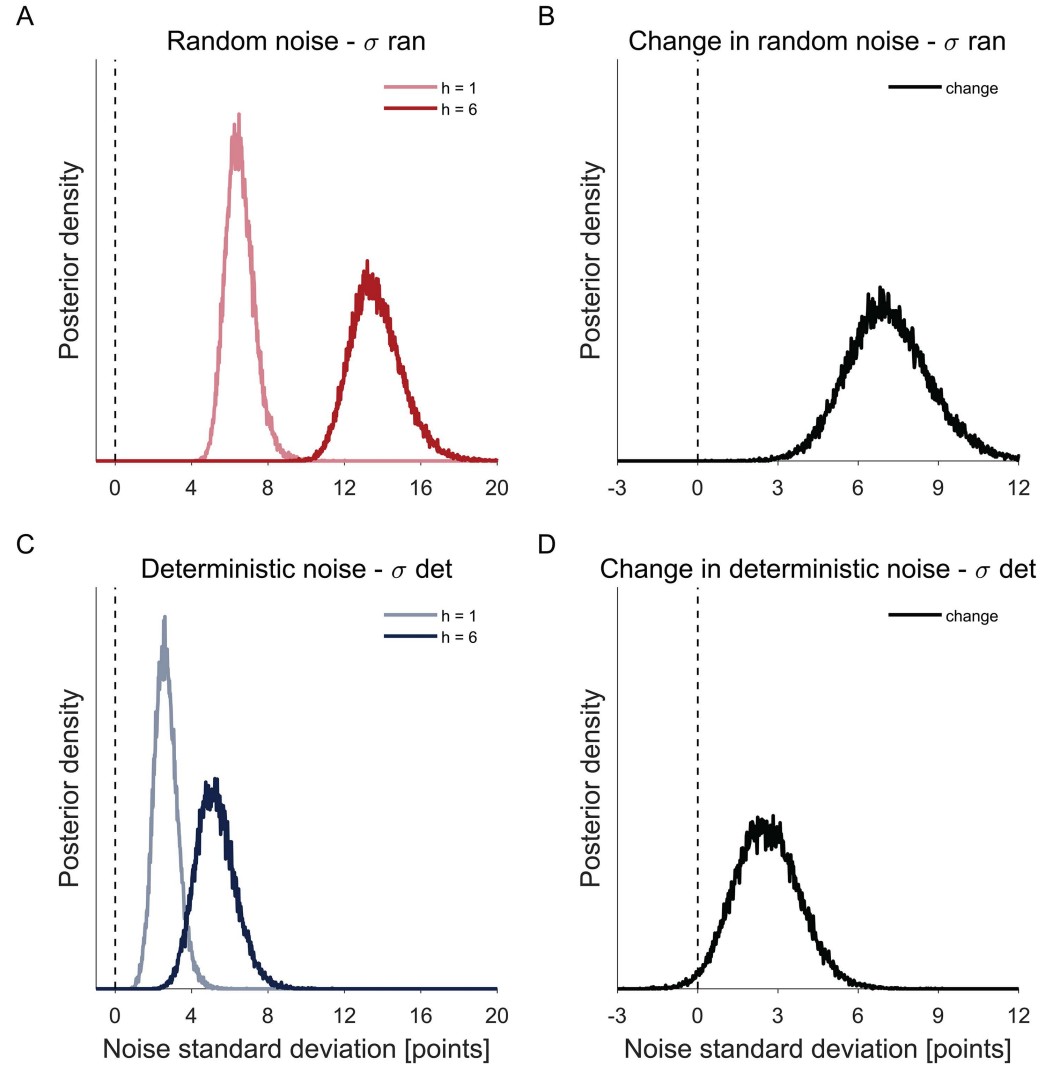

**Fig 5. Model based analysis showing the posterior distributions over the group-level mean of the standard deviations of random and deterministic noise.** Both random (A, B) and deterministic (C,D) noises are nonzero (A, C) and increase with horizon **(B, D)**.

it is very similar to the relative increase in random noise with horizon $\sigma^{ran}_{horizon6}/\sigma^{ran}_{horizon1}$. Similar results are found in other variants of our model (S12 and S13 Figs).

To ensure that the joint increase of random and deterministic noises is genuine and not an artifact from the fitting procedure, we computed the correlation between ground-truth values of random noise, and best-fitting values of deterministic noise (and vice versa), and they do not correlate (S14 Fig). Furthermore, we simulated data from a series of reduced models with known random and deterministic noise values in which either random or deterministic noise does not change with Horizon, and fit our model to the simulated data. Our model detects and only detects a change in random/deterministic noise with horizon, when the change is present in the model that simulates the data (S15 and S16 Figs, section 2.2 in S1 Text).

**Posterior predictive checks.** In addition to fitting the model to behavior, it is also important to check whether the model captures the qualitative patterns of the data [22,23] — specifically how p(high info), p(low mean) and p(inconsistent) change with horizon.

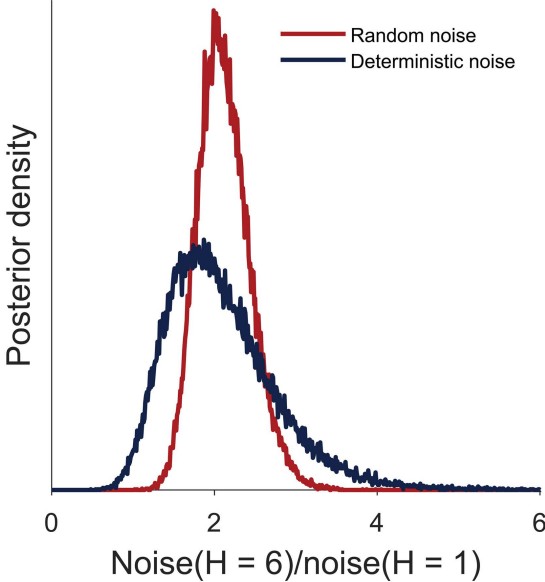

**Fig 6. Model based analysis showing the posterior distributions over the ratio of the group-level mean of the standard deviations of random and deterministic noise between horizon 6 and horizon 1 respectivelly.** The ratios in the standard deviations of noises between horizon 6 and horizon 1 are similar for random and deterministic noise.

To perform this 'posterior predictive check', we created a set of simulated data by taking the subject-level parameters from the hierarchical Bayesian fits and having the model play the same sequence of games as seen by the subjects. We then applied the same model-free analysis as described in the previous sections to this simulated data set and compared the model's behavior to that of participants. As shown in Fig 7, the model can account for all qualitative patterns in the data — the increase in p(high info), p(low mean), and p(inconsistent) with horizon, and that p(inconsistent) is in between pure random and pure deterministic noise. The quantitative agreement is almost perfect for p(high info) and for p(inconsistent), but the model slightly overestimate p(low mean) in [2 2] conditions. This has to do with the skewness of the subject-level posterior distribution (see section 2.3 in S1 Text).

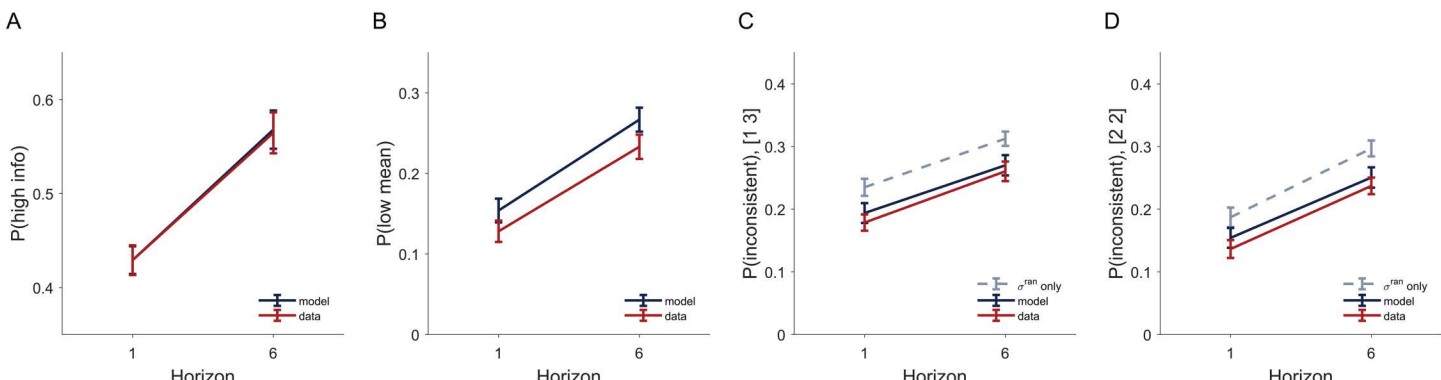

**Fig 7. Our model accounts for all qualitative patterns of the data, namely, (A) p(high info) and (B) p(low mean) increase as a function of horizon, p(inconsistent) increases as a function of horizon for both [1 3] (C) and [2 2] (D) conditions and it lies between the pure random and pure deterministic noise prediction.**

As a control, we also applied posterior predictive checks on alternative models that consider only deterministic or only random noise, and these reduced models fail to capture all qualitative patterns (S17, S18 Figs). Full details of this analysis can be found in section 2.3 in S1 Text.

## Discussion

In this paper, we investigated whether random exploration is really random or whether it is driven deterministically by aspects of the stimulus we have previously ignored when measuring 'decision noise'. Using a version of the Horizon Task with repeated games, we found evidence that at least some of the noise in random exploration could be explained by such 'deterministic noise'. In particular, we found that deterministic noise accounted for around 14% of the overall variability in people's behavior.

One interpretation for this low level of deterministic noise is that most of the variability in random exploration is truly random. Such a random noise interpretation, would be consistent with recent work showing that variability in perceptual decisions may be driven by imperfections in mental inference [24]. In this view, apparently random behavior is not due to sensory processing or response selection, but to suboptimal computations in the brain. Although suboptimal inference is different from simply adding random noise to neural circuitry [25], as long as the suboptimality in neural computation is not a deterministic function of the stimuli, it is a form of random noise in our definition. Indeed, a strong interpretation of this hypothesis would suggest that randomness in explore-exploit behavior is due to imperfect inference about the correct course of action. In the context of the Horizon Task, such computational errors would likely be larger in the long horizon condition as the correct course of action in these cases is much harder to compute [26].

Although the random noise interpretation is theoretically appealing, our approach, while an improvement on previous methods, is not without limitations. Most important is that our measure of 'random' noise is only an upper bound on the true level of randomness and that, in principle, the random decision noise could be lower. Specifically, in our model, what we labeled random noise was really 'non-stimulus-driven variability'. While this non-stimulus-driven variability could be driven by truly random stochastic processes, it could also be driven by deterministic processing that is unrelated to the stimuli in the task. For example, such deterministic noise could be driven by differences in where people look, or for how long they look, or by whether they were fidgeting or scratching their nose [27]. Another limitation is that deterministic noise is defined based on the stimulus within a game, between-game deterministic strategies and stimuli from previous games (e.g., memory of previously seen game) were also treated as random noises in our model, although our model could be potentially extended by considering deterministic noise over a sequence of stimuli across games. In addition to this conceptual limitation in measuring deterministic noise, parameter recovery simulations suggest that our estimation method also slightly underestimates deterministic noise (see Figs 4 and S5-S8). As a result, from both a conceptual and methodological perspective, it is possible that the remaining 86% of the decision noise that is not stimulus-driven noise, could be deterministic.

Like the random noise account, the deterministic noise account is also in line with previous work in which neural variability can be accounted for by fluctuations in sensory inputs. For example, MT neurons were shown to have a reproducible temporal modulation in response to a fixed random motion stimulus [28]. In other words, 'irrelevant' features in the stimuli are represented in a reliable way in the brain that could drive downstream choices in a predictable way.

Regardless of whether we interpret the noise as random or deterministic, a key finding in this paper is that both types of noise change with horizon. Such a horizon increase is a hallmark of an exploratory process and suggests that the modulation of deterministic and random processes may underlie random exploration. Moreover, the fact that the horizon change in the two types of noise are proportional to each other (Fig 6) suggests a possible mechanism for random exploration: a reduction in the strength with which reward drives the choice.

We show first that a change in noise is mathematically equivalent to a change in reward signal strength in our decision model (see also [29]). To show how a change in reward processing could affect random and deterministic noise, consider

the simple decision model we introduced in Equation 2. In this model, choice is determined by the sign of the difference in utility $\Delta Q$ between the two options, where

$$\Delta Q = \Delta R + A\Delta I + b + n_{det} + n_{ran} \tag{3}$$

Now imagine a case where the reward signal is scaled by a factor $\beta$. In this case, $\Delta Q$ becomes

$$\Delta Q = \beta\Delta R + A\Delta I + b + n_{det} + n_{ran} \tag{4}$$

Because the choice only depends on the sign of $\Delta Q$, scaling $\Delta Q$ by a factor of $1/\beta$ will not change the behavior of the model. Thus, if we divide both sides of the above equation by $\beta$ we get

$$\Delta Q/\beta = \Delta R + A\Delta I/\beta + b/\beta + n_{det}/\beta + n_{ran}/\beta \tag{5}$$

which is equivalent to a scaling of both deterministic and random noise by the same factor $1/\beta$. Thus, one interpretation of our result that both deterministic and random noise change across horizons with the same ratio, is that this reflects a change in reward processing. That is, the reward signal is reduced in the longer horizon condition (smaller $\beta$ in horizon 6 than horizon 1).

Such a reduction in the strength of reward coding in exploration, is consistent with our recent work using a drift diffusion model (DDM) to model explore-exploit decisions [30]. In the drift diffusion model, changes in behavioral variability can be driven by changes in the decision threshold (smaller threshold = more noise) or changes in the signal-to-noise ratio with which reward is encoded (lower SNR = more noise). By fitting both choices and response times, we were able to distinguish between these two accounts showing that the majority of the horizon-change in variability was driven by changes in SNR and not threshold. However, this model could not determine whether the changes in SNR were driven by signal or noise. By showing that the change in deterministic and random noise have approximately the same ratio, the present work suggests that this SNR change is driven by changes in reward-signal processing, not noise. Of course, to truly see whether changes in signal or noise are driving random exploration will require more direct measurements of neural processing such as with neuroimaging and electrophysiology [31–34].

## Materials and methods

### Ethics statement

Human subject protocols were approved by the University of Arizona institutional review board (IRB # 1411567117). Written informed consent was given by all participants prior to participating in the study.

### Participants

80 participants (ages 18–25, 37 male, 43 female) from the University of Arizona undergraduate subject pool participated in the experiment. 15 were excluded on the basis of performance, using the same exclusion criterion as in [12]. In this exclusion criteria, we measured the accuracy of each participant's choices by calculating the percentage of times that a participant chose the bandit with the higher underlying mean payouts in the last choice of a long horizon game, intuitively people should figure out which bandit has a higher mean payout by the last trial and should have an accuracy measure significantly above 50%, specifically, we computed the likelihood that the measured accuracy can be achieved by making a completely random choice between the two options and excluded participants with a likelihood greater than 0.1%, in other words, participants who didn't show an accuracy significant above chance with $p < 0.001$ were excluded in the analysis. This left 65 for the main analysis. Note that including the 15 badly performing subjects did not change the main results (S1, S2, and S11 Figs).

## Task

The task was a modified version of the Horizon Task [12] (Fig 1). In this task, participants play a set of games in which they make choices between two slot machines (one-armed bandits) that pay out rewards from different Gaussian distributions. In each game they made multiple decisions between two options. Each option paid out a random reward between 1 and 100 points sampled from a Gaussian distribution. The means of the underlying Gaussians were different for the two bandit options, remained the same within a game, but changed with each new game. One of the bandits always had a higher mean than the other. Participants were instructed to maximize the points earned over the entire task. To maximize their rewards in each game, participants need to exploit the slot machine with the highest mean, but they cannot identify this best option without exploring both options first.

The number of games participants played depended on how well they performed, which acted as the primary incentive for performing the task. Thus, the better participants performed, the sooner they got to leave the experiment. On average, participants played 153.7 games (minimum = 90 games, maximum = 192 games) and the whole task lasted between 12.37 and 32.15 minutes (mean 22.78 minutes). Participants played an average of 65.3 repeated pairs of games (minimum = 30 repeated pairs, maximum = 79 repeated pairs).

As in the original paper [12], the distributions of payoffs tied to bandits were independent between games and drawn from a Gaussian distribution with variable means and fixed standard deviation of 8 points. Differences between the mean payouts of the two slot machines were set to either 4, 8, 12 or 20. One of the means was always equal to either 40 or 60 and the second was set accordingly. Participants were informed that in every game one of the bandits always has a higher mean reward than the other. The order of games was randomized. Mean sizes and order of presentation were counterbalanced.

Each game consisted of 5 or 10 choices. Every game started with a fixation cross, then a bar of boxes appeared indicating the horizon for that game. For the first 4 trials - the instructed 'forced-choice' trials, we highlight the box on one of the bandits to instruct the participant to choose that option. On these trials, they have to press the corresponding key to reveal the outcome. From the fifth trial, boxes on both bandits will be highlighted and they are free to make their own decision. There was no time limit for decisions. During free choices participants could press either the left arrow key or right arrow key to indicate their choice of left or right bandit. The score feedback was presented for 300ms. The task was programmed using Psychtoolbox in MATLAB [35,36].

The first four trials of each game were forced-choice trials, in which only one of the options was available for the participant to choose. We used these forced-choice trials to manipulate the relative ambiguity of the two options, by providing the participant with different amounts of information about each bandit before their first free choice. The four forced-choice trials set up two uncertainty conditions: unequal uncertainty(or [1 3]) in which one option was forced to be played once and the other three times, and equal uncertainty(or [2 2]) in which each option was forced to be played twice. After the forced-choice trials, participants made either 1 or 6 free choices (two horizon conditions).

## Model-based analysis

We modeled behavior on the first free choice of the Horizon Task using a version of the logistic choice model in [12] that was modified to differentiate deterministic noise from random noise. Because the stimuli are identical in the repeated games, by definition, deterministic noise remains the same in repeated games, whereas random noise can change.

**Hierarchical Bayesian model.** To model participants' choices on this first free-choice trial, we assume that they make decisions by computing the difference in value $\Delta Q$ between the right and left options, choosing right when $\Delta Q > 0$ and left otherwise. Specifically, we write

$$\Delta Q = \Delta R + A\Delta I + b + n_{det} + n_{ran} \tag{6}$$

where, the experimentally controlled variables are $\Delta R = R_{right} - R_{left}$, the difference between the mean of the rewards shown on the forced trials, and $\Delta I$, the difference of information available for playing the two options on the first free-choice trial. For simplicity, and because information is manipulated categorically in the Horizon Task, we define $\Delta I$ to be +1, -1 or 0, +1 if one reward is drawn from the right option and three are drawn from the left in the [1 3] condition, -1 if one from the left and three from the right, and in [2 2] condition, $\Delta I$ is 0. The other variables are: the spatial bias, $b$, which determines the extent to which participants prefer the option on the right; the information bonus $A$, which controls the level of directed exploration; $n_{det}$ and $n_{ran}$ are deterministic noise and random noise respectively. $n_{det}$ denotes the deterministic noise, which is identical on the repeat versions of each game; and $n_{ran}$ denotes random noise, which is uncorrelated between repeat plays and changes every game.

Each subject's behavior in each horizon condition is described by 4 free parameters (Table 1): the information bonus $A$, the spatial bias, $b$, the standard deviation of the deterministic noise, $\sigma_{det}$, and the standard deviation of the random noise, $\sigma_{ran}$. Each of the free parameters is fit to the behavior of each subject using a hierarchical Bayesian approach [37]. In this approach to model fitting, each parameter for each subject is assumed to be sampled from a group-level prior distribution whose parameters, the so-called 'hyperparameters', are estimated using a Markov Chain Monte Carlo (MCMC) sampling procedure (Fig 8). The hyper-parameters themselves are assumed to be sampled from 'hyperprior' distributions whose parameters are defined such that these hyperpriors are broad.

Table 1. Model parameters, priors, hyperparameters and hyperpriors.

| Parameter | Prior | Hyperparameters | Hyperpriors |
|---|---|---|---|
| information bonus, $A_{is}$ | $A_{is} \sim \text{Gaussian}(\mu_i^A, \sigma_i^A)$ | $\theta_i^A = (\mu_i^A, \sigma_i^A)$ | $\mu_i^A \sim \text{Gaussian}(0, 100)\ \sigma_i^A \sim \text{Exponential}(0.01)$ |
| spatial bias, $b_{is}$ | $b_{is} \sim \text{Gaussian}(\mu_i^b, \sigma_i^b)$ | $\theta_i^b = (\mu_i^b, \sigma_i^b)$ | $\mu_i^b \sim \text{Gaussian}(0, 100)\ \sigma_i^b \sim \text{Exponential}(0.01)$ |
| deviation of deterministic noise, $\sigma_{isg}^{det}$ | $\sigma_{is}^{det} \sim \text{Gamma}(k_i^{det}, \lambda_i^{det})$ | $\theta_i^{det} = (k_i^{det}, \lambda_i^{det})$ | $k_i^{det} \sim \text{Exponential}(0.01)\ \lambda_i^{det} \sim \text{Exponential}(10)$ |
| deviation of random noise, $\sigma_{isgr}^{ran}$ | $\sigma_{is}^{ran} \sim \text{Gamma}(k_i^{ran}, \lambda_i^{ran})$ | $\theta_i^{ran} = (k_i^{ran}, \lambda_i^{ran})$ | $k_i^{ran} \sim \text{Exponential}(0.01)\ \lambda_i^{ran} \sim \text{Exponential}(10)$ |

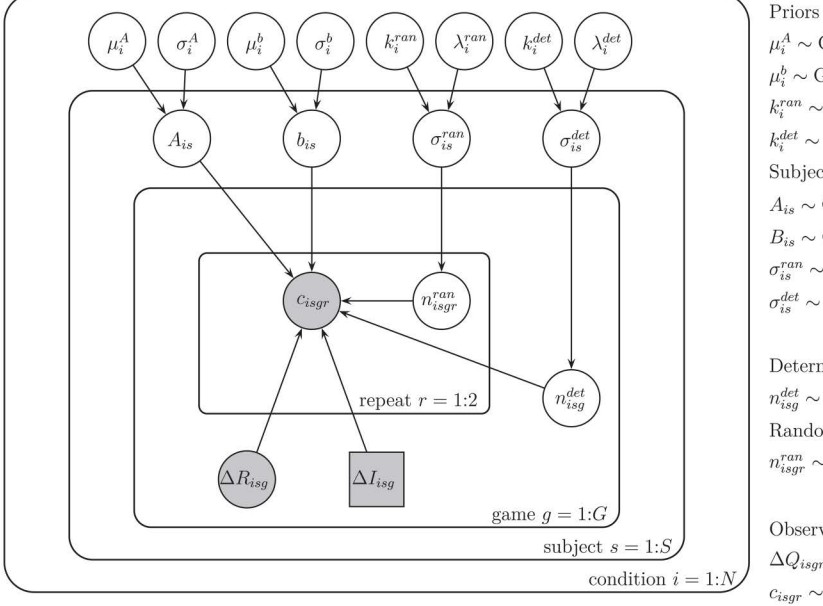

Priors
$\mu_i^A \sim \text{Gaussian}(0, 100),\ \sigma_i^A \sim \text{Exponential}(0.01)$
$\mu_i^b \sim \text{Gaussian}(0, 100),\ \sigma_i^b \sim \text{Exponential}(0.01)$
$k_i^{ran} \sim \text{Exponential}(0.01),\ \lambda_i^{ran} \sim \text{Exponential}(10)$
$k_i^{det} \sim \text{Exponential}(0.01),\ \lambda_i^{det} \sim \text{Exponential}(10)$
Subject specific parameters
$A_{is} \sim \text{Gaussian}(\mu_i^A, \sigma_i^A)$
$B_{is} \sim \text{Gaussian}(\mu_i^B, \sigma_i^B)$
$\sigma_{is}^{ran} \sim \text{Gamma}(k_i^{ran}, \lambda_i^{ran})$
$\sigma_{is}^{det} \sim \text{Gamma}(k_i^{det}, \lambda_i^{det})$

Deterministic noise for repeated game
$n_{isg}^{det} \sim \text{Logistic}(0, \sigma_{is}^{det})$
Random noise for each game
$n_{isgr}^{ran} \sim \text{Logistic}(0, \sigma_{is}^{ran})$

Observed choices
$\Delta Q_{isgr} \leftarrow \Delta R_{isg} + A_{is}\Delta I_{isg} + b_{is} + n_{isgr}^{ran} + n_{isg}^{det}$
$c_{isgr} \sim \text{Bernoulli}(Q_{isgr} > 0)$

**Fig 8. Schematic of the hierarchical Bayesian model using notation of [38].**

The particular priors and hyperpriors for each parameter are shown in Table 1. For example, we assume that the information bonus, $A^{is}$, for each horizon condition $i$ and for each participant $s$, is sampled from a Gaussian prior with mean $\mu_i^A$ and standard deviation $\sigma_i^A$. These prior parameters are sampled in turn from their respective hyperpriors: $\mu_i^A$, from a Gaussian distribution with mean 0 and standard deviation 10, and $\sigma_i^A$ from an Exponential distribution with parameters 0.1.

**Model fitting using MCMC.** The model was fit to the data using Markov Chain Monte Carlo approach implemented in the JAGS package [20] via the MATJAGS interface (psiexp.ss.uci.edu/research/programs_data/jags). This package approximates the posterior distribution over model parameters by generating samples from this posterior distribution given the observed behavioral data.

In particular we used 10 independent Markov chains to generate 50000 samples from the posterior distribution over parameters (5000 samples per chain). Each chain had a burn in period of 5000 samples, which were discarded to reduce the effects of initial conditions, and posterior samples were acquired at a thin rate of 1. Convergence of the Markov chains was confirmed *post hoc* by eye.

## Supporting information

**S1 Fig. Replication of previous findings with data from all participants (i.e., no exclusions).** (A) model-free measure of behavioral variability, $p(\text{low mean})$, increases with horizon. (B) model-free measure of information seeking, $p(\text{high info})$, increases with horizon. (C) model-based measure of behavioral variability, decision noise $\sigma$, increases with horizon. (D) model-based measure of information seeking, information bonus $A$, increases with horizon.
(TIFF)

**S2 Fig. Model-free analysis with data from all participants (i.e., no exclusions) suggests that both deterministic and random noise contribute to the choice variability in random exploration.** For both the [1 3] (A) and [2 2] (B) condition, people show greater choice inconsistency in horizon 6 than horizon 1. However, the extent to which their choices are inconsistent lies between what is predicted by purely deterministic and random noise, suggesting that both noise sources influence the decision.
(TIFF)

**S3 Fig. Model-free analysis with simulated choices from a model that has only random noise validates our prediction of p(inconsistent) for pure random noise.** The extent to which simulated choices are inconsistent completely overlaps with our pure random noise prediction($p > 0.05$). This suggests that when choice inconsistency lies below the pure random noise prediction indeed provides evidence that deterministic noise exists in random exploration (Fig 3).
(TIFF)

**S4 Fig. Deterministic noise can recover known deterministic processes that's intentionally omitted by the model.** In the reduced model where the deterministic effect of uncertainty condition is omitted from the model, deterministic noise is higher compared to the full model that accounts for the effect of uncertainty. Random noise remains unchanged between the two models.
(TIFF)

**S5 Fig. Hyperprior recovery.** Parameter recovery over the posterior distribution of random and deterministic noise standard deviations $\sigma_{det}$ and $\sigma_{ran}$. Solid lines are true posterior used to simulate choices. Lighter color shades represent the re-fitted posterior to the simulated choices. Our model fitting procedure faithfully recovers the non-stimulus-driven random noise (A, B), but systematically underestimates deterministic noise in both horizons (D, E). The horizon differences in random noise is also faithfully recovered (C). The horizon differences in deterministic noise is also underestimated but not significant (F).
(TIFF)

**S6 Fig. Frequentist coverage analysis.** Parameter recovery over the mean estimates of random and deterministic noise standard deviations $\sigma_{det}$ and $\sigma_{ran}$. Solid lines are true posterior used to simulate choices, dashed black line is the mean of the true posterior. Histograms represent the mean estimates of the respective parameters in the refitting to the simulated data. (A) and (B) are random noise at H = 1 and H = 6, respectively. (C) is the random noise differences between horizons. (D) and (E) are deterministic noise at H = 1 and H = 6, respectively. (F) is the deterministic noise differences between horizons.
(TIFF)

**S7 Fig. Parameter recovery.** Parameter recovery over the subject-level means of information bonus, *A*, spatial bias, *b*, random noise standard deviation, $\sigma_{ran}$, and deterministic noise standard deviation, $\sigma_{det}$, for horizon 1 (left column) and horizon 6 (right column) games.
(TIFF)

**S8 Fig. Parameter recovery (200 repetitions).** Same as S7 Fig, except that the recovered parameters were averaged across 200 repetitions and then compared to the original parameters.
(TIFF)

**S9 Fig. Parameter recovery with 0 random noise or 0 deterministic noise.** Parameter recovery over the posterior of random noise standard deviation, $\sigma_{ran}$, and deterministic noise standard deviation, $\sigma_{det}$, for purely random noise (top row) and purely deterministic noise (bottom row) games.
(TIFF)

**S10 Fig. Parameter recovery on arbitrary combinations of random and deterministic noises.** A. Recovered posterior distributions of random noise. B. Recovered posterior distributions of deterministic noise. For both A and B, from the top row to the bottom row, the true noise standard deviation that is used in the simulations go from 0 to 10. The y limit of each panel is 4 (+/- 2 from the true value). Our model did a relatively good job in recovering all combinations of deterministic and random noises.
(TIFF)

**S11 Fig. Model based analysis with data from all participants (i.e., no exclusions) showing the posterior distributions over the group-level mean of the standard deviations of random and deterministic noise.** Both random (A, B) and deterministic (C,D) noises are nonzero (A, C) and change with horizon (B, D). However, random noise has both a greater magnitude overall (A, C) and a greater change with horizon (B, D) than deterministic noise.
(TIFF)

**S12 Fig. Model based analysis from a model that estimates random and deterministic noises separately for [1 3] and [2 2] conditions.** The posterior distributions over the group-level mean of the standard deviations of random and deterministic noise. Both random (A, E) and deterministic (C,G) noises are nonzero (A, C, E, G) and change with horizon (B, D, F, H). However, random noise has both a greater magnitude overall (A, E) and a greater change with horizon (B, F) than deterministic noise. Moreover, both random and deterministic noises have a greater magnitude in [1 3] compared to [2 2] conditions.
(TIFF)

**S13 Fig. Model based analysis from a model that uses variance differences as dl.** The posterior distributions over the group-level mean of the standard deviations of random and deterministic noise. Both random (A, B) and deterministic (C,D) noises are nonzero (A, C) and change with horizon (B, D). However, random noise has both a greater magnitude overall (A, C) and a greater change with horizon (B, D) than deterministic noise.
(TIFF)

**S14 Fig. Parameter recovery for shuffled data.** To show that the joint increase of random and deterministic sources of noise is not caused by a limitation of the fitting procedure, we calculated the correlation between ground-truth values of random noise, and best-fitting values of deterministic noise (and vice versa). Ground-truth values are shuffled best-fit parameters. As expected, ground-truth random values do not correlate with recovered deterministic noises, showing that the increase of deterministic noise with horizon is genuine and not a by-product of increase of random noise, and vice versa.
(TIFF)

**S15 Fig. Model based analysis with reduced models.** Each row is one model. These models varied in whether deterministic $\sigma^{det}$ and random noise $\sigma^{ran}$ are present or not and whether either types of noise is dependent on horizon (subscript denotes the dependence on horizon).
(TIFF)

**S16 Fig. Hyperprior recovery of reduced models.** Our model qualitatively captures whether deterministic and random noise are present or not and whether either types of noise is dependent on horizon. A-D. both deterministic and random noise are horizon dependent, E-H. only random noise is horizon dependent, I-L. only deterministic noise is horizon dependent, M-P. neither random nor deterministic noise is horizon dependent, Q-T. only deterministic noise is assumed to be present, U-X. only random noise is assumed to be present.
(TIFF)

**S17 Fig. Posterior checks for reduced models Model comparison.** A-D. both deterministic and random noise are horizon dependent, E-H. only random noise is horizon dependent, I-L. only deterministic noise is horizon dependent, M-P. neither random nor deterministic noise is horizon dependent, Q-T. only deterministic noise is assumed to be present, U-X. only random noise is assumed to be present.
(TIFF)

**S18 Fig. Posterior checks for reduced models (maximal likelihood estimation) Model comparison (using maximal likelihood estimation).** A-D. both deterministic and random noise are horizon dependent, E-H. only random noise is horizon dependent, I-L. only deterministic noise is horizon dependent, M-P. neither random nor deterministic noise is horizon dependent, Q-T. only deterministic noise is assumed to be present, U-X. only random noise is assumed to be present.
(TIFF)

**S1 Text. Supplemental Materials containing additional analyses and results.**
(PDF)

## Author contributions

**Conceptualization:** Siyu Wang, Robert C. Wilson.

**Data curation:** Siyu Wang.

**Formal analysis:** Siyu Wang.

**Funding acquisition:** Robert C. Wilson.

**Investigation:** Siyu Wang.

**Methodology:** Siyu Wang, Robert C. Wilson.

**Project administration:** Robert C. Wilson.

**Resources:** Robert C. Wilson.

**Software:** Siyu Wang.

**Supervision:** Robert C. Wilson.

**Visualization:** Siyu Wang.

**Writing – original draft:** Siyu Wang, Robert C. Wilson.

**Writing – review & editing:** Siyu Wang, Robert C. Wilson.

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
