## [Decision Letter · Decision Letter 0]

18 Oct 2024

Dear Wang,

Thank you very much for submitting your manuscript "Separating random and deterministic sources of computational noise in explore-exploit decisions" for consideration at PLOS Computational Biology.

As with all papers reviewed by the journal, your manuscript was reviewed by members of the editorial board and by several independent reviewers. In light of the reviews (below this email), we would like to invite the resubmission of a significantly-revised version that takes into account the reviewers' comments.

We cannot make any decision about publication until we have seen the revised manuscript and your response to the reviewers' comments. Your revised manuscript is also likely to be sent to reviewers for further evaluation.

Sincerely,

Andrea E. Martin, Ph.D.

Section Editor

PLOS Computational Biology

Reviewer's Responses to Questions

**Comments to the Authors:**

Reviewer #1: This manuscript describes a cognitive modeling study of human choice behavior in a well-known task (the Horizon task) developed by the last author. In this study, the authors ask whether part of the decision noise that is captured by the standard cognitive model used to fit human choices is not genuinely random (unpredictable), but rather deterministic (predictable). To split decision noise in the Horizon task into random and deterministic terms, the authors apply a repeated-trial approach that has been described and applied in tightly connected contexts to show that at least 14% of random exploration is due to deterministic biases that affect choices in the same way across repetitions of the same trial. More interestingly, the authors report that not only random, but also deterministic sources of noise increase with horizon length, an effect that the authors discuss in terms of a decrease in the decision gain of the reward term (rather than an increase in the decision gain of the information bonus term).

I found the manuscript to be very well written and structured in terms of analyses and results, the authors adequately motivate their study in the introduction and the (repeated-trial) approach they have chosen to use to address their research question. The methods also appear well suited to provide statistical support for their findings, and their discussion of the results (in particular the joint increase of random and deterministic noise components with horizon length) is interesting from a cognitive perspective. I nevertheless have a few comments below which the authors should address in my opinion to make the manuscript stronger and better reflect the existing literature that has used in recent years the exact same approach - in very similar contexts - to decompose choice variability into random and deterministic components.

* The current version of the manuscript is missing recent references (copied below) that describe (ref. 1 and 2) and apply/discuss (ref. 3 and 4) the same (repeated-trial) approach to cognitive problems that are tightly connected to the one studied here. These references should be cited in the revised manuscript to provide additional theoretical (and empirical) background about this bias-variance separation approach. They would also provide additional findings that can be used to discuss the findings obtained by the authors in the present study:

1/ Wyart V, Koechlin E (2016) Choice variability and suboptimality in uncertain environments. Current Opinion in Behavioral Sciences 11, 109-115. doi:10.1016/j.cobeha.2016.07.003

2/ Wyart V (2018) Leveraging decision consistency to decompose suboptimality in terms of its ultimate predictability. Behavioral and Brain Sciences 41, e248. doi:10.1017/S0140525X18001504 - Commentary on Rahnev D, Denison RN (2018) Suboptimality in perceptual decision making. Behavioral and Brain Sciences 41, e223. doi:10.1017/S0140525X18000936

3/ Findling C, Skvortsova V, Dromnelle R, Palminteri S, Wyart V (2019) Computational noise in reward-guided learning drives behavioral variability in volatile environments. Nature Neuroscience 22(12), 2066-2077. doi:10.1038/s41593-019-0518-9

4/ Findling C, Wyart V (2021) Computation noise in human learning and decision-making: origin, impact, function. Current Opinion in Behavioral Sciences 38, 124-132. doi:10.1016/j.cobeha.2021.02.018

* The authors have applied parameter recovery and posterior predictive checks to check whether their fitting procedure is capable of estimating parameter values, and of predicting the key features of the observed human choice behavior. These two procedures have been described as critically important by Wilson and Collins (2019, eLife) and by Palminteri, Wyart and Koechlin (2017, Trends in Cognitive Sciences - missing reference which should ideally be cited where posterior predictive checks are first described). They reveal that the deterministic noise term is underestimated by the fitting procedure, and that the best-fitting model overestimates p(low mean) and p(inconsistent) in the [2,2] condition - which is the most ‘basic’ condition without information bonus.

The authors do not discuss these biases of the fitting procedure, but it would be important to understand why it is the case. Could it be due to the hierarchical fitting approach used by the authors? Why did the authors choose this hierarchical fitting approach over and above a simpler, independent (subject-wise) fitting approach? The theoretical merits of a hierarchical fitting approach are clear and obvious, but could the authors employ the simpler subject-wise fitting approach to check that the same biases of the fitting procedure remain present (and therefore that they are not triggered by the hierarchical fitting approach)?

* The authors appear to take for granted that the joint increase of random and deterministic sources of noise with horizon length is genuine and not caused by a limitation of the fitting procedure. I tend to agree with their interpretation, but it would be very useful to provide additional empirical evidence in the main text that this joint increase is indeed genuine. The parameter recovery approach illustrated in Figure 4 should include panels of figures found in the Supplementary Materials which show that the fitting procedure is capable of correctly recovering arbitrary combinations of random and deterministic sources of noise. A simpler and more compact test, which the authors should perform and plot as a new panel of Figure 4, would be to plot the confusion matrix arising from the parameter recovery procedure (Figure 4 only shows what corresponds to the diagonal of the confusion matrix). It is indeed critically important that simulated ground-truth values of random noise do not correlate with best-fitting values of deterministic noise (and vice versa).

This would require to fit choice behavior using a non-hierarchical, subject-wise fitting approach, or to complexify the hierarchical fitting approach to estimate the covariance between random and deterministic sources of noise. Both control analyses would be valid, and I let the authors choose whichever approach they find most appropriate for their data. This would provide empirical evidence (already available to the authors since they have performed a parameter recovery analysis) that random and deterministic sources of noise can indeed be reliably separated, and therefore that the joint increase of the two forms of noise with horizon length is genuine. This type of control analysis have been performed in a recent study, in case this is helpful:

Lee JK, Rouault M, Wyart V (2023) Adaptive tuning of human learning and choice variability to unexpected uncertainty. Science Advances 9, add0501. doi:10.1126/sciadv.add0501

Reviewer #2: In the manuscript entitled 'Separating random and deterministic sources of computational noise in explore-exploit decisions', Siyu Wang and Robert C. Wilson present an extension of the Horizon task by Wilson and colleagues (2014) to investigate whether random exploration in human decision-making is driven by stochastic processes in the brain or by some unobserved deterministic process. The task is extended so as to disentangle deterministic noise from random noise by presenting a situation where, unbeknownst to them, participants are presented with the exact same choice twice. This enables the authors to estimate a lower bound on the amount of variability that is deterministically driven by the stimulus and an upper bound on the amount of variability that is random. They found evidence that at least 14% of the variability in random exploration in their task can be accounted for by deterministic processing of the stimulus.

The topic of this research is very interesting, timely and of importance to the community. The maths behind the computational models and the analyses seem correct and are elegantly developed. But at the end of the reading I am left only partially satisfied. The starting important question is: where does the 'random' noise identified by Wilson et al. 2014 come from? But the article arrives to the conclusion that there's around 14% of explainable random noise on a task where there are important limitations to the protocol (see detailed comments below), and without really finding explanations of where do those 14% of deterministic noise come from, not why participants' choices tend to be repeated. This would require to look at when random (or deterministic) choices occur. For example, when the average rewards displayed by the bandits are close (this is taken into account in the model), when the uncertainty displayed on one of the two bandit arms is higher than for the other (risk seeking or risk averse behavior), when there is an attractor point (a higher reward) for the choice that is sub-optimal. The authors themselves admit this in the discussion: 'As a result, from both a conceptual and methodological perspective, it is possible that the remaining 86% of the decision noise that is not stimulus-driven noise, could be deterministic. ' I wish they'd develop more hypotheses on this.

Overall, I feel that some control analyses and ways to discard alternative interpretations are needed.

ANALYSES

I am intrigued by the negative information bonus A in the model-based analyses for Horizon=1 (Figure 2D). Is it significantly different from 0? If yes, does this mean that participants are even avoiding uncertainty (risk aversiveness) in that case? What would be the implications of this?

I do not fully understand how the plotted values for the 'pure random noise prediction' (i.e., 'random noise only' in the figures) and 'deterministic noise only' in Fig. 3, Suppl. Fig. S2 and S3 were computed. If these correspond to theoretical values for the choice inconsistency for the purely deterministic and purely random noise cases, as formalized pages 11 and 12, then I don't understand why these values have standard deviations and vary so much between figures: in the [2 2] condition of Figure 3, the plotted mean of random noise only are <0.2 for Horizon 1 and <0.3 for Horizon 6, while in the [2 2] condition of Suppl. Fig. S3, they are >0.2 for Horizon 1 and >0.3 for Horizon 6. I expect that the simulated data in the [2 2] condition of Suppl. Fig. S3 are significantly different from the 'pure random noise prediction' (i.e., 'random noise only') in the [2 2] condition of Figure 3. Isn't it a problem and shouldn't the authors solve it here?

I think the model validation analyses in supplementary data are very useful and well-performed. Nevertheless, shouldn't the spatial bias term be kept in all model versions to make them comparable? Could the authors quantitatively show how much the spatial bias term contributes to explaining participants' behavior? In Suppl. Fig. S7, it seems difficult to recover the spatial bias parameter with the parameter recovery method. Why is that so?

Moreover, it would be useful to give the reader a quick grasp of the summarized results by showing a model recovery matrix (as in Wilson & Collins 2019) with all nested versions of the full model (those in Table S1). Does the full model win when the simulations are generated by the full model? Does a reduced model without random noise win when the simulations are generated by the very same model? Conversely, does a reduced model without deterministic noise win when the simulations are generated by the very same model?

Do I understand correctly that in the full model as well as in the reduced one, the random noise (sigma ran) and the deterministic noise (sigma det) come into play only during the first choices that are repeated during two games, and not during other first choices nor during 2nd, 3rd, etc. choices? Or instead are these two noise terms contributing to all decisions? Could the authors make this clearer in the manuscript, please?

Could the authors add a few sentences in the supplementary information to clarify that if the random noise increases in the reduced model (the one without information bonus), it leads to less choice consistency between repeated games, and that conversely if the deterministic noise increases, it leads to more choice consistency?

Does the deterministic noise in the reduced model capture and replace the effect that the information bonus produces in the full model?

The ‘posterior predictive check’ analysis is very nice and still rarely performed in the literature. Nevertheless, why are the model and data so different for p(low mean) (Figure 7B)? Is the difference significant? How can this be explained and interpreted?

I'll go into more details about risk averse and risk seeking behavior. In the model, discretizing information amounts is an approximation that may have important consequences in terms of explanatory power and interpretation of the results. In particular, setting a value of 0 for information in a situation [2 2] is bit unsatisfying: even if the four elements have been drawn from distributions with the same variance, we may have two values close together on the left and two far apart on the right. For example, if you have two identical values on one side and two different values on the other, it's very clear that there's information to be found on one side but not on the other. This is not taken into account in the model and falls under the heading of 'random' noise, which sounds somehow a bit absurd. Changing this in the model wouldn't require much effort. For example, replacing -1, 0, 1 by the variance differences. The variance of a single element is 0, so in [1 3] and [3 1] we go from -1 or 1 to the value of the variance of the 3 elements, with a plus or minus sign in front. And in the case [2 2] we go from 0 to a difference in variance that is probably small but potentially non-zero. This wouldn't change the model much, but it would directly change the interpretation (what if the 14% came from there?). Otherwise, it gives me the impression that the authors are a bit over-interpreting their results of a model that may not be ideal, and that the value of 14% doesn't mean much.

There is a contradiction between the sentence 'these reduced models fail to capture all qualitative patterns (Supplementary Figure S13)' in the main article, and the sentence 'As shown in Figure S13, only one of these alternative models, where random noise is horizon dependent but deterministic noise is not, can capture the full qualitative pattern of behavior.' in Supplementary Information. The main original should clarify that one of the alternative models does capture qualitative patterns. Moreover, the authors should clarify which quantitative measure they used to decide 'not as good' in sentence 'However, the quantitative fit to the data is not as good (Figure S13)' in Supplementary Information. To me it seems in Suppl. Fig. S13 that the difference between the two models (Fig. S13 A-D vs. Fig. S13 E-H) and the data is not significant. Again, I think that a model recovery analysis would be needed here.

In the Discussion section, the demonstrations of how a change in reward processing could affect random and deterministic noise should be acknowledged as similar to the demonstration of how a change in reward processing could affect random noise in Cinotti et al. 2019 Scientific Reports, where it is written that when 'all Q-values are downscaled in the same proportion as the reward [and] When these values are plugged into the softmax process, the result is exactly equivalent to a decrease of the inverse temperature, again in the same proportion.'

INTERPRETATION

What if the participants had sometimes a good memory of having already been confronted with the same game + the bandit they had previously chosen, and deterministically decide to pick the other bandit so at to see which payout they obtain in this case? This would be deterministic exploration policy at the game level rather than at the bandit level, in contrast to the authors' interpretation as non-stimulus-driven random noise in explore-exploit decisions. The authors argue that two identical games in their task are 'separated by several minutes in time so as to avoid detection'. But how could they unsure that the repetition has never been detected? What would be the interpretation of their results if let's say at least a proportion of game repetitions had been detected by the participants? It seems to me that a way to address this problem would be to redo the task and ask 2 questions after each game's first free choice: have you already encountered the same game before? If yes, which bandit had you chosen the previous time? On the one hand, this would prompt people to know that there are game repetitions, which would increase their vigilance towards this feature and would increase their detection probability, on the other hand, this would enable to separate repeated games for which participants' accurately remembered their previous choice from those where they failed to remember. Another solution, so as not to bias participants' responses during the task would be to ask them to fill a questionnaire after, where they are asked whether they think they have encountered twice the same situation. This would be less ideal than the proposed variant of the task, but at least give an idea whether this was a problem or not in the present task. Have the authors asked the participants such questions during a post-task questionnaire?

LITERATURE

In the abstract, I suggest to replace 'recent work suggests that variability can actually be adaptive' by 'a long body of machine learning work suggests that variability can actually be adaptive'.

In the introduction, to be fair with the existing computational literature on adaptive decision noise, after the sentence 'It has recently been shown that humans appear to use random exploration and can increase decision noise when it is more beneficial to explore (Findling et al., 2019, Gershman, 2018, Wilson et al., 2014)', I suggest the authors add the following: , as has also been suggested in computational models of animal behavior (Doya 2002 Neural Networks; Khamassi et al., 2013 Progress in Brain Research).

Typos

Page 8, a logistic distributions.

Page 12, in the both the.

Page 19, a fixed random motion stimuli -> stimulus.

Page 25, in the reference by Beck et al. 2012, there is a duplication of bibliographical information. Same thing for Findling et al., Tomov et al., Musall et al., Hogeveen et al., Ebitz et al., and Costa et al.

Page 8 of Suppl. Info. (Section 2.3) to recovery parameters -> to recover.

**Have the authors made all data and (if applicable) computational code underlying the findings in their manuscript fully available?**

Reviewer #1: **No:** The authors report that the data and code will be made available upon publication.

Reviewer #2: Yes

PLOS authors have the option to publish the peer review history of their article (what does this mean? ). If published, this will include your full peer review and any attached files.

Reviewer #1: No

Reviewer #2: **Yes:** Mehdi Khamassi
---

## [Decision Letter · Decision Letter 1]

15 Feb 2026

Dear Wang,

We are pleased to inform you that your manuscript 'Separating random and deterministic sources of computational noise in explore-exploit decisions' has been provisionally accepted for publication in PLOS Computational Biology.

Best regards,

Jean Daunizeau

Academic Editor

PLOS Computational Biology

Andrea E. Martin

Section Editor

PLOS Computational Biology

Reviewer's Responses to Questions

**Comments to the Authors:**

Reviewer #1: I apologize to the authors for my late re-review of the manuscript. I have read the revised manuscript and the authors’ responses to all comments made by the two reviewers. I found that the authors have provided convincing responses to all comments, and I am happy to support publication of the revised manuscript without further revisions. I only add comments to two specific responses of the comments that were the most important in my earlier review:

Response 1.2

I understand the authors’ response regarding the rationale behind using hierarchical fits in this case. The figures are flipped in terms of the x and y axis labels I think: it is the recovered estimates that are very large, not the simulated values for random and deterministic noise parameters I think.

I am happy with the authors’ updates in the manuscript regarding the fact that their fitting method provides lower bounds on deterministic noise levels.

Response 1.3

The response is perfectly convincing and conclusive, and strengthens the reported findings of joint increases in random and deterministic noise components with horizon length.

I was also convinced by the authors’ careful responses to Reviewer #2. I congratulate the authors for an insightful study.

Reviewer #2: The authors have substantially improved their control analyses and the clarity of their manuscript. Congratulations for this rigorous and stimulating work!

**Have the authors made all data and (if applicable) computational code underlying the findings in their manuscript fully available?**

Reviewer #1: Yes

Reviewer #2: Yes

PLOS authors have the option to publish the peer review history of their article (what does this mean? ). If published, this will include your full peer review and any attached files.

**Do you want your identity to be public for this peer review?** For information about this choice, including consent withdrawal, please see our Privacy Policy .

Reviewer #1: No

Reviewer #2: **Yes:** Mehdi Khamassi